# Blind Video Temporal Consistency via Deep Video Prior

**Chenyang Lei**[*]     **Yazhou Xing**[*]     **Qifeng Chen**
The Hong Kong University of Science and Technology

## Abstract

Applying image processing algorithms independently to each video frame often leads to temporal inconsistency in the resulting video. To address this issue, we present a novel and general approach for blind video temporal consistency. Our method is only trained on a pair of original and processed videos directly instead of a large dataset. Unlike most previous methods that enforce temporal consistency with optical flow, we show that temporal consistency can be achieved by training a convolutional network on a video with the Deep Video Prior. Moreover, a carefully designed iteratively reweighted training strategy is proposed to address the challenging multimodal inconsistency problem. We demonstrate the effectiveness of our approach on 7 computer vision tasks on videos. Extensive quantitative and perceptual experiments show that our approach obtains superior performance than state-of-the-art methods on blind video temporal consistency. Our source codes are publicly available at `github.com/ChenyangLEI/deep-video-prior`.

## 1   Introduction

Numerous image processing algorithms have demonstrated great performance in single image processing tasks [9, 15, 23, 38, 41], but applying them directly to videos often results in undesirable temporal inconsistency (e.g., flickering). To encourage video temporal consistency, most researchers design specific methods for different video processing tasks [20, 25, 28] such as video colorization [21], video denoising [24] and video super resolution [32]. Although task-specific video processing algorithms can improve temporal coherence, it is unclear or challenging to apply similar strategies to other tasks. Therefore, a generic framework that can turn an image processing algorithm into its video processing counterpart with strong temporal consistency is highly valuable. In this work, we study a novel approach to obtain a temporally consistent video from a processed video, which is a video after independently applying an image processing algorithm to each frame of an input video.

Prior work has studied general frameworks instead of task-specific solutions to improve temporal consistency [3, 7, 19, 39]. Bonneel et al. [3] present a general approach that is blind to image processing operators by minimizing the distance between the output and the processed video in the gradient domain and a warping error between two consecutive output frames. Based on this approach, Yao et al. [39] further leverage more information from a key frame stack for occluded areas. However, these two methods assume that the output and processed videos are similar in the gradient domain, which may not hold in practice. To address this issue, Lai et al. [19] maintain the perceptual similarity with processed videos by adopting a perceptual loss [16]. In addition to blind video temporal consistency methods, Eilertsen et al. [7] propose a framework to finetune a convolutional network (CNN) by enforcing regularization on transform invariance if the pretrained CNN is available. Moreover, most approaches [3, 19, 39] enforce the regularization based on dense correspondence (e.g., optical flow or PatchMatch [1]), and the long-term temporal consistency often degrades.

---

[*]Joint first authors

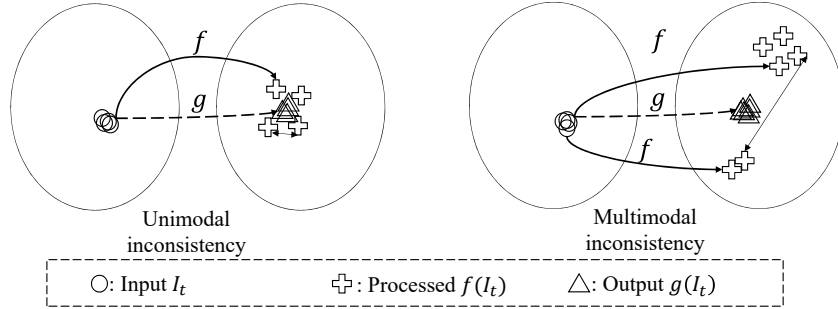

Figure 1: Illustration of unimodal and multimodal inconsistency. In unimodal inconsistency, similar input video frames are mapped to moderately different processed frames within the same mode by $f$. In multimodal inconsistency, similar input video frames may be mapped to processed frames within two or more modes by $f$. A function $g$ is to improve the temporal consistency for these two cases.

We propose a general and simple framework, utilizing the Deep Video Prior by training a convolutional network on videos: the outputs of CNN for corresponding patches in video frames should be consistent. This prior allows recovering most video information first before the flickering artifacts are eventually overfitted. Our framework does not enforce any handcrafted temporal regularization to improve temporal consistency, while previous methods are built upon enforcing feature similarity for correspondences among video frames [3, 19, 39]. Our idea is related to DIP (Deep Image Prior [37]), which observes that the structure of a generator network is sufficient to capture the low-level statistics of a natural image. DIP takes noise as input and trains the network to reconstruct an image. The network performs effectively to inverse problems such as image denoising, image inpainting, and super-resolution. For instance, the noise-free image will be reconstructed before the noise since it follows the prior represented by the network. We conjecture that the flickering artifacts in a video are similar to the noise in the temporal domain, which can be corrected by deep video prior.

Our method only requires training on the single test video, and no training dataset is needed. Training without large-scale data has been adopted commonly in internal learning [34, 37]. In addition to DIP [37], various tasks [8, 33, 35, 40] show that great performance can be achieved by using only test data.

As another contribution, we further propose a carefully designed iteratively reweighted training (IRT) strategy to address the challenging multimodal inconsistency problem. Multimodal inconsistency may appear in a processed video. Our method selects one mode from multiple possible modes to ensure temporal consistency and preserve perceptual quality. We apply our method to diverse computer vision tasks. Results show that although our method and implementation are simple, we do not only show better temporal consistency but also suffer less performance degradation compared with current state-of-the-art methods.

## 2  Background

Let $I_t$ be the input video frame at time step $t$ and the corresponding processed frame $P_t = f(I_t)$ can be obtained by applying the image processing algorithm $f$. For instance, $f$ can be image colorization, image dehazing, or any other algorithm. Blind video temporal consistency [3, 19] aims to design a function $g$ to generate a temporally consistent video $\{O_t\}_{t=1}^{T}$ for $\{P_t\}_{t=1}^{T}$.

**Temporal inconsistency.** Temporal inconsistency appears when the same object has inconsistent visual content in $\{P_t\}_{t=1}^{T}$. For example, in Fig. 5, some corresponding local patches of two input frames vary a lot in processed frames. There are mainly two types of temporal inconsistency in a processed video: unimodal inconsistency and multimodal inconsistency. The left schematic diagram in Fig. 1 illustrates unimodal inconsistency, in which the $\{I_t\}_{t=1}^{T}$ are consistent and $\{P_t\}_{t=1}^{T}$ are not that consistent (e.g., flickering artifacts). For some tasks, multiple possible solutions exist for a single input (e.g., for colorization, a car might be colorized to red or blue). As a result, the temporal inconsistency in $\{P_t\}_{t=1}^{T}$ is visually more obvious, as shown in the right schematic diagram in Fig. 1.

**Correspondence-based regularization.** Previous work [3, 19] usually uses correspondences to improve temporal consistency: correspondences via optical flow or PatchMatch [1] in two frames

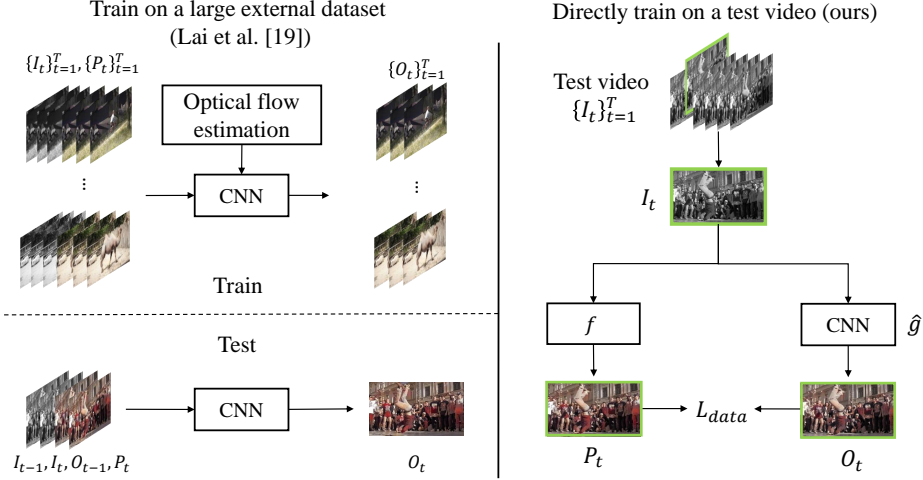

Figure 2: The overview of our framework. The previous learning-based method [19] requires a large-scale dataset which consists of input frames $\{I_t\}_{t=1}^T$ and processed frames $\{P_t\}_{t=1}^T$ pairs to train the network. Different from this, our pipeline directly trains on a test video. The network $\hat{g}$ is trained to mimic the image operator $f$. Note that network $\hat{g}$ is not a specific type of network so that our pipeline can adapt to different tasks. For each iteration, only one frame is used for training.

should share similar features (e.g., color or intensity). A regularization loss $L_{reg}$ is defined to minimize the distance between correspondences in the output frames $\{O_t\}_{t=1}^T$. Also, a reconstruction loss $L_{data}$ is used to minimize distance between $\{O_t\}_{t=1}^T$ and $\{P_t\}_{t=1}^T$. Therefore, a loss function [19] or objective function [3] $L$ is commonly adopted for blind video temporal consistency:

$$L = L_{data} + L_{reg}, \tag{1}$$

$$L_{reg} = \sum_{t=2}^T \|O_t - W(O_{t-1}, F_{t\rightarrow t-1})\|, \tag{2}$$

where $F_{t\rightarrow t-1}$ is the optical flow from $I_t$ to $I_{t-1}$ and $W$ is the warping function. Note that our model does not need the $L_{reg}$ and thus avoid optical flow estimation.

## 3 Method

### 3.1 Deep Video Prior (DVP)

While previous work [3, 7, 19] designs $L_{reg}$ in various ways, we claim $L_{reg}$ can be implicitly achieved by Deep Video Prior (DVP): the outputs of CNN for corresponding patches in video frames should be consistent. This prior is based on an observation and a fact: the outputs of a CNN on two similar patches are expected to be similar at the early stage of training; the same object in different video frames has similar appearances. The DVP allows recovering most video information while eliminating flickering before eventually overfitting to all information including inconsistency artifacts.

As shown in Fig. 2, we propose to use a fully convolutional network $\hat{g}(\cdot; \theta)$ to mimic the original image operator $f$ while preserving temporal consistency. Different from Lai et al. [19], only a single video is used for training $\hat{g}$, and only a single frame is used in each iteration. We initialize $\hat{g}$ randomly, and then it can be optimized in each iteration with a single data term without any explicit regularization:

$$\arg\min_\theta \ L_{data}(\hat{g}(I_t; \theta), P_t), \tag{3}$$

where $L_{data}$ measures the distance (e.g., $L_1$ distance) between $\hat{g}(I_t; \theta)$ and $P_t$. We stop training when $\{O_t\}_{t=1}^T$ is close to $\{P_t\}_{t=1}^T$ and before artifacts (e.g., flickering) are overfitted. Through this basic architecture, unimodal inconsistency can be alleviated. A neural network $\hat{g}$ and a data term $L_{data}$ are yet to design in our framework. In practice, we adopt U-Net [30] and perceptual loss [6, 16]

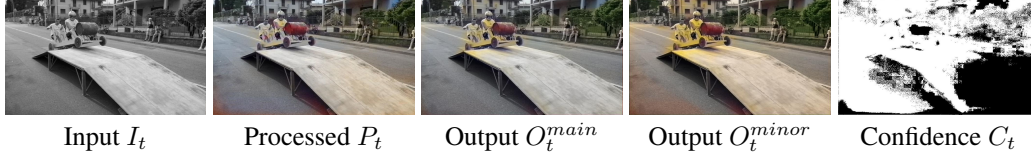

| Input $I_t$ | Processed $P_t$ | Output $O_t^{main}$ | Output $O_t^{minor}$ | Confidence $C_t$ |

Figure 3: A confidence map can be calculated to exclude outliers.

for all evaluated tasks. Note that $\hat{g}$ is not restricted to U-Net, other appropriate CNN architectures (e.g., FCN [26] or original architecture of $f$) are also applicable.

**Analysis.** We analyze a toy example in Fig. 4 to study deep video prior. Consider 8 consecutive video frames as input $\{I_t\}_{t=1}^8$, and the processed frames $\{P_t\}_{t=1}^8$ are obtained by adding noise on ground truth. Two kinds of inconsistency are synthesized. The first row is illustrated for unimodal inconsistency: $\{P_t\}_{t=1}^8$ are close to each other but have small distance; the second and third rows are illustrated for multimodal inconsistency: $\{P_t\}_{t=1}^8$ are separated to two clusters where the distance between two clusters is large. In Fig. 4(a), in the beginning, i.e. 100-th iteration, the outputs for consecutive frames are highly overlapped, which means $\{O_t\}_{t=1}^8$ are consistent with each other. At 200-th iteration, they begin to separate from each other and the distance between $\{O_t\}_{t=1}^8$ becomes larger. After 1000 iterations, $\{O_t\}_{t=1}^8$ are not consistent anymore and are quite similar with $\{P_t\}_{t=1}^8$. For unimodal inconsistency at 100-th iteration, $\{O_t\}_{t=1}^8$ are both close to ground truth and consistent with each other. In practice, we adopt this phenomenon to solve unimodal inconsistency problem. However, for multimodal inconsistency shown in Fig. 4(b), although $\{O_t\}_{t=1}^8$ is consistent at 100-th iteration, they are not close to either ground truth. To address this problem, we propose an IRT strategy in Section 3.2. In Fig. 4(c), after applying our IRT, we can obtain the results which are both consistent and close to one ground truth at a stage (100-th and 200-th iteration).

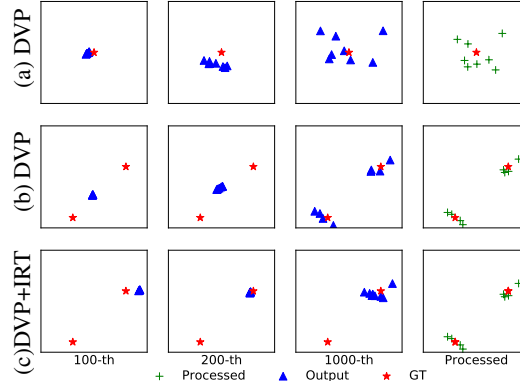

Figure 4: A toy example that demonstrates deep video prior. (a): Training by DVP on $\{P_t\}_{t=1}^8$ with unimodal inconsistency. (b): Training by DVP on $\{P_t\}_{t=1}^8$ with multimodal inconsistency. (c): Training by DVP and IRT on $\{P_t\}_{t=1}^8$ with multimodal inconsistency. In the beginning, the outputs $\{O_t\}_{t=1}^8$ tend to be consistent with each other. After 200 iterations, outputs $\{O_t\}_{t=1}^8$ tend to separate from each other and completely overfit the processed frames $\{P_t\}_{t=1}^8$ after 1000 iterations. In cooperation with DVP and IRT strategy, we can handle the challenging multimodal inconsistency.

## 3.2 Iteratively Reweighted Training (IRT)

We propose an iteratively reweighted training (IRT) strategy for multimodal inconsistency because it cannot be solved by our basic architecture easily. Since the difference between multiple modes can be quite large, averaging different modes can result in a poor performance, which is far from any ground-truth mode, as shown in Fig. 1 and Fig. 4. As a result, previous methods fail to generate consistent results [19] or tend to degrade the original performance largely [3].

In IRT, a confidence map $C_t \in \{0,1\}^{H \times W \times 1}$ is designed to choose one main mode for each pixel from multiple modes and ignore the outliers (one minor mode or multiple modes). We calculate the pixel-wise confidence map $C_{t,i}$ based on the network $\hat{g}(I; \theta^i)$ trained at $i$-th iteration. We increase the number of channels in the network output (e.g., six channels for two RGB images) to obtain two outputs: a main frame $O_{t,i}^{main}$ and an outlier frame $O_{t,i}^{minor}$. The confidence map $C_{t,i}$ (i.e., confidence map for main mode) can then be calculated by:

$$C_{t,i}(x) = \begin{cases} 1, & d(O_{t,i}^{main}(x), P_t(x)) < \max\{d(O_{t,i}^{minor}(x), P_t(x)), \delta\}, \\ 0, & otherwise \end{cases} \quad (4)$$

where $d$ is the function to measure the distance between pixels and $\delta$ is a threshold. We use $L_1$ distance as $d$ and set $\delta$ to 0.02. For a pixel $x$, if the $P_t(x)$ is closer to $O_{t,i}^{main}$ compared with the other modes $O_{t,i}^{minor}$ at $i$-th iteration, the confidence will be 1. For some pixels, only a single mode exists. Hence, the distance to both $O_{t,i}^{main}$ and $O_{t,i}^{minor}$ can be small and these pixels should be selected for training. As shown in Fig. 3, for pixels that are: (1) close to both $O_{t,i}^{main}$ and $O_{t,i}^{minor}$; (2) closer to $O_{t,i}^{main}$, the confidence is high. The function of confidence maps is similar to cluster assignment in K-Means [27] when K=2 and the pixels in minor mode are similar to the outliers in iteratively reweighted least squares (IRLS) [11]. At last, in the $(i+1)$-th iteration, the training loss function can be updated by the confidence map:

$$\theta^{i+1} = \arg\min_{\theta} \quad L_{data}(C_{t,i} \odot O_{t,i}^{main}, C_{t,i} \odot P_t) +$$
$$L_{data}((1 - C_{t,i}) \odot O_{t,i}^{minor}, (1 - C_{t,i}) \odot P_t). \qquad (5)$$

In practice, we can use a specific frame (e.g., the first frame) to train the network for the main mode at the beginning of training. By doing so, we can make sure that the main outputs are close to the specific mode.

## 4 Experiments

We first evaluate our framework through the 7 tasks in our experiments.

**Colorization.** A gray image can be colorized through the single image colorization algorithm [14, 41]. Multimodal inconsistency appears when a gray video is processed since there are multiple possible colors for one gray input.

**Dehazing.** Single image dehazing aims at removing haze to recover the clear underlying scenes in an image. However, directly applying an image dehazing algorithm (e.g., He et al. [10]) to videos might lead to high-frequency inconsistency artifacts.

**Image enhancement.** We utilize DBL [9] as the image enhancement algorithm. Their results reveal high-frequency flickering artifacts in videos.

**Style transfer.** Though image style transfer (e.g., WCT [22]) has achieved excellent performance, applying style transfer to videos is quite challenging because new content is generated in each frame.

**Image-to-image translation.** When applying the pretrained CycleGAN [42] model for image-to-image translation on videos, inconsistent artifacts appear due to newly generated textures.

**Intrinsic decomposition.** Intrinsic decomposition [2] aims at decomposing each image $I$ into reflectance $R$ and shading $S$, which satisfies $I = R \times S$. Following Lai et al. [19], $R$ and $S$ are processed, respectively. The inconsistency in this task is relatively large.

**Spatial white balancing.** White balance is a crucial task to eliminate color casts due to differing illuminations. When applying a single image white balance algorithm [12] to videos, we notice that multimodal inconsistency appears.

### 4.1 Experimental setup

**Baselines.** We mainly choose two state-of-the-art methods [3, 19] whose source codes or test results are available. Some characteristics of baselines are listed in Table 1. Compared with previous approaches [3, 19], our method is simple since we do not need training datasets or estimating optical flow. In addition to the simplicity, temporal consistency, and data fidelity of our method are also quite competitive.

|  | Training dataset | Optical flow | Temporal consistency | Data fidelity |
|---|---|---|---|---|
| [3] | **No** | Yes | **Good** | Moderate |
| [19] | Yes | Yes | Moderate | **Good** |
| Ours | **No** | **No** | **Good** | **Good** |

Table 1: Comparisons among our method and Bonnel et al. [3], Lai et al. [19]. Comparisons of temporal consistency and data fidelity are based on evaluation results in Section 4.3.

**Datasets.** Following the previous work [3, 19], we adopt the DAVIS dataset [29] and the test set

| Task | $E_{warp} \downarrow$ | | | | $F_{data} \uparrow$ | | |
|---|---|---|---|---|---|---|---|
| | Processed | [3] | [19] | Ours | [3] | [19] | Ours |
| Dehazing [10] | 0.139 | 0.150 | 0.149 | **0.127** | 24.67 | 24.79 | **30.44** |
| Spatial White Balancing [12] | 0.158 | 0.149 | 0.147 | **0.139** | 21.80 | 24.70 | **27.80** |
| Colorization/ Iizuka et al. [14] | 0.181 | **0.170** | 0.173 | 0.174 | 26.50 | **30.65** | 30.19 |
| Colorization/ Zhang et al. [41] | 0.187 | **0.172** | 0.180 | 0.175 | 24.31 | **29.90** | 28.20 |
| Enhancement [9]/ expertA | 0.197 | **0.176** | 0.183 | 0.179 | 23.98 | **27.77** | 25.77 |
| Enhancement [9]/ expertB | 0.188 | 0.177 | 0.180 | **0.175** | 25.23 | 28.46 | **28.81** |
| CycleGAN [42]/ ukiyoe | 0.224 | 0.161 | 0.164 | **0.159** | 22.04 | **26.90** | 25.09 |
| CycleGAN [42]/ vangogh | 0.215 | **0.192** | 0.202 | 0.193 | 21.17 | **26.53** | 25.80 |
| Intrinsic [2]/ reflectance | 0.211 | **0.160** | 0.176 | 0.164 | 21.75 | 24.37 | **24.97** |
| Intrinsic [2]/ shading | 0.204 | 0.158 | 0.175 | **0.152** | 21.75 | 23.48 | **24.61** |
| Style Transfer [22]/ antimo. | 0.280 | 0.242 | 0.253 | **0.235** | 15.89 | 24.14 | **24.43** |
| Style Transfer [22]/ candy | 0.277 | 0.242 | 0.251 | **0.234** | 14.93 | **23.53** | 22.47 |
| Average Score | 0.2051 | 0.1791 | 0.1860 | **0.1755** | 22.00 | 26.27 | **26.55** |

Table 2: Our method achieves comparable numerical performance compared with Bonneel et al. [3] and Lai et al. [19]. Note that these metrics do not completely reflect the visual quality of output videos.

collected by Bonneel et al. [3] for evaluation. The test set collected by Bonneel et al. [3] is adopted for dehazing and spatial white balancing. The DAVIS dataset [29], which contains 30 videos, is used for the rest applications. Since our model is trained on test data directly, we do not need to collect the training set.

**Implementation details.** We use the Adam optimizer [18] and set the learning rate to 0.0001 for all the tasks. The batch size is 1. Dehazing, spatial white balancing, and image enhancement are trained for 25 epochs. Intrinsic decomposition, colorization, style transfer, and CycleGAN are trained for 50 epochs.

## 4.2 Evaluation metrics

**Temporal inconsistency.** The warping error $E_{warp}$ is used to measure the temporal inconsistency. For each frame $O_t$, we calculate the warping error with frame $O_{t-1}$ [5, 13, 19] and the first frame $O_1$ [17] for considering both short-term and long-term consistency. The final $E_{warp}$ is calculated by:

$$E_{pair}(O_t, O_s) = \frac{1}{\sum_{i=1}^{N} M_{t,s}(x_i)} \sum_{i=1}^{N} M_{t,s}(x_i) ||O_t(x_i) - W(O_s)(x_i)||_1, \qquad (6)$$

$$E_{warp}(\{O_t\}_{t=1}^{T}) = \frac{1}{T-1} \sum_{t=2}^{T} \{E_{pair}(O_t, O_1) + E_{pair}(O_t, O_{t-1})\}, \qquad (7)$$

where $M_{t,s}$ is the occlusion map [31] for a pair of images $O_t$ and $O_s$, $N$ is the number of pixels, and $W$ is backward warping with optical flow [36].

**Performance degradation.** Avoiding performance degradation is critical to blind temporal consistency. Since we do not have the ground truth videos for most evaluated tasks, we use data fidelity $F_{data}$ between $\{P_t\}_{t=1}^{T}$ and $\{O_t\}_{t=1}^{T}$ as a reference to evaluate the performance degradation. Note that data fidelity does not mean the perceptual performance of a video. For example, for $\{P_t\}_{t=1}^{T}$ with multimodal inconsistency, high-quality results $\{O_t\}_{t=1}^{T}$ have only a single mode, i.e., the distance can be quite large for some frames, as shown in Fig. 5. Since the first frame is used as a reference in baselines [3, 19], the first frame is excluded for fair comparison:

$$F_{data}(\{P_t\}_{t=1}^{T}, \{O_t\}_{t=1}^{T}) = \frac{1}{T-1} \sum_{t=2}^{T} PSNR(P_t, O_t). \qquad (8)$$

## 4.3 Results

**Quantitative results.** We show the quantitative comparison results in Table 2. We obtain the best average scores at both temporal consistency and data fidelity metrics. Although Bonneel et al. [3]

| | Colorization | CycleGAN | Enhancement | Intrinsic | WhiteBalance | StyleTransfer | Dehazing | Average |
|---|---|---|---|---|---|---|---|---|
| | [14, 41] | [42] | [10] | [9] | [12] | [22] | [2] | |
| Processed | 7% | 6% | 12% | 3% | 0% | 1.5% | 5% | 5% |
| [3] | 36% | 24% | 19.5% | 14% | 16% | 23.5% | 25% | 23% |
| [19] | 16.5% | 10.5% | 34% | 18.5% | 6% | **40.5%** | 5% | 18.71% |
| Ours | **40.5%** | **59.5%** | **34.5%** | **64.5%** | **78%** | 34.5% | **65%** | **53.79%** |

Table 3: On average, our results are significantly preferred in the user study.

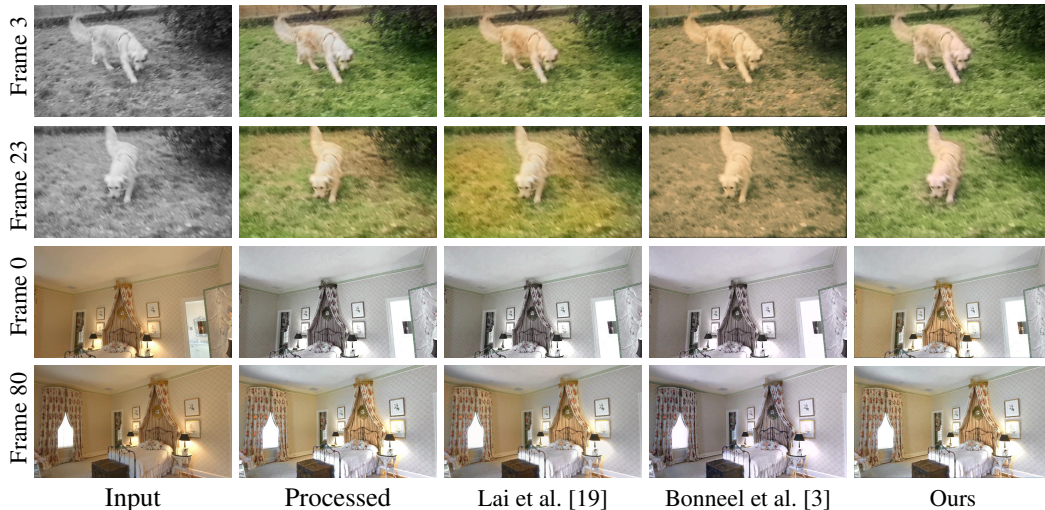

Figure 5: The input frames are processed by colorization [41] and white balancing [12] respectively. As shown in these two examples, the results of Lai et al. [19] look similar to the processed video but fail to preserve long-term consistency. Also, the results by Bonneel et al. [3] have obvious performance decay: the color is changed in an undesirable way. Our method solves the multimodal inconsistency with IRT by producing a video with long-term consistency for the main mode.

also achieve similar temporal consistency, they suffer from severe performance decay problems in some tasks, as shown in Table 2. Lai et al. [19] obtain comparable results for data fidelity metrics, but our temporal consistency performance is much better. We also observe that they cannot enforce long term consistency for multimodal cases while their data fidelity performs well, and an example is illustrated in Fig. 5.

**Qualitative results.** More perceptual results will be presented in supplementary material due to limited space. Readers are recommended to check videos for better perceptual evaluation. Our performance is much better than baselines to solve multimodal inconsistency problem. As shown in Fig. 5, the processed frames have two completely different results. Our framework produces temporally consistent results and also maintains the performance in one mode. Lai et al. [19] fail to impose reasonable temporal regularization on both cases. The results of Bonneel et al. [3] suffer from the serious performance decay problem. These qualitative results are consistent with our quantitative results. In Fig. 8, we compute the mean intensity of an image to evaluate the temporal consistency. The flickering artifact of processed frames is quite obvious. For Lai et al. [19], the flickering artifacts are handled well in the short term, but the difference between the first and last frame is too large. Also, although the amplitude of flickering is decreased by Bonneel et al. [3], flickering still exists. Compared with baselines, our result is consistent in both the short term and long term.

**User study.** We conduct a user study to compare the perceptual preference of various methods. In total, 107 videos from all tasks are randomly selected. 20 subjects are asked to select the best videos with both temporal consistency and performance similarity. As shown in Table 3, our method outperforms baselines [3, 19] and processed videos in most tasks. For tasks with simple unimodal inconsistency like enhancement [9], the difference between our method and Lai et al. [19] is minor. However, for tasks with multimodal inconsistency, our method can outperform theirs to a large extent.

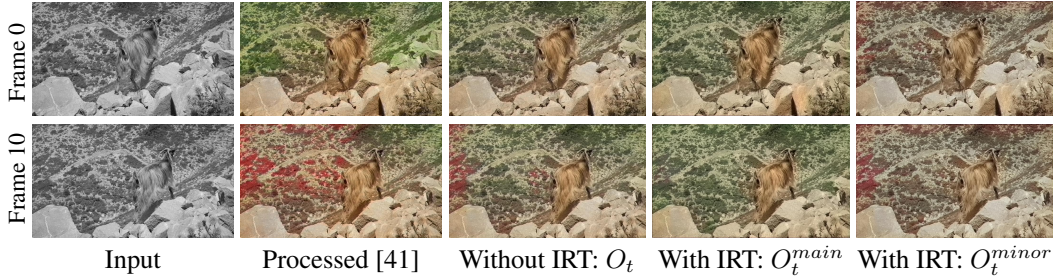

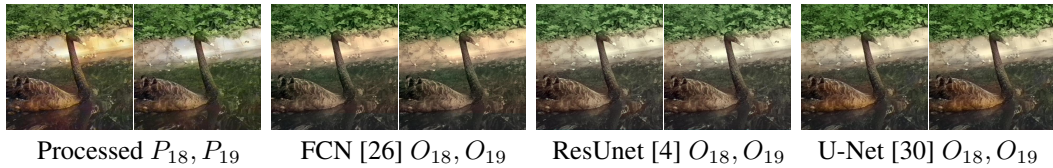

| Input | Processed [41] | Without IRT: $O_t$ | With IRT: $O_t^{main}$ | With IRT: $O_t^{minor}$ |

Figure 6: With IRT, our method can improve video temporal consistency for the multimodal case, compared with the basic training architecture.

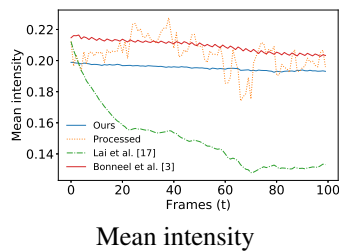
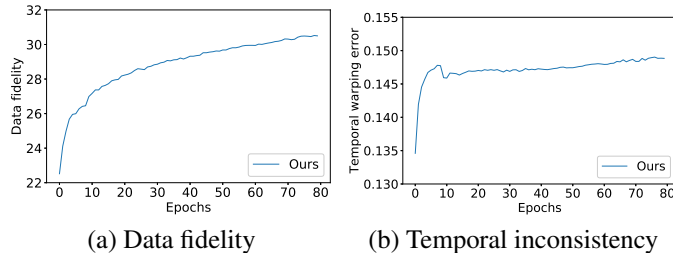

Processed $P_{18}, P_{19}$     FCN [26] $O_{18}, O_{19}$     ResUnet [4] $O_{18}, O_{19}$     U-Net [30] $O_{18}, O_{19}$

Figure 7: Deep video prior is applied to various CNN architectures, and thus the temporal consistency of results by different CNNs is consistently better than processed frames [41].

Mean intensity     (a) Data fidelity     (b) Temporal inconsistency

Figure 8: Mean intensity of frames on dehazing [10].

Figure 9: Both temporal inconsistency $E_{warp}$ and data fidelity $F_{data}$ are increasing in the process of training.

## 4.4 Analysis

**Ablation study.** An ablation study is conducted to analyze the importance of IRT. We implement another version that removes IRT for colorization [41] on the DAVIS dataset [29]. We find that $F_{data}$ of basic architecture is higher: 29.61 for basic architecture and 28.20 for IRT. However, the perceptual performance with IRT is more consistent, as shown in Fig. 6. This phenomenon is not surprising since it follows our motivation in Section 3.2.

Another controlled experiment is conducted to verify our framework by two more architectures: FCN [26] and ResUnet [4]. As shown in Fig. 7, the deep video prior also can be applied to these two networks. Although there is a minor difference among results of three architectures, the temporal consistency of each architecture is satisfying compared with processed frames. More comparisons will be presented in the supplementary material.

**When to stop training?** In our method, there is a trade-off between temporal consistency and data fidelity. This phenomenon is reported in all previous methods [3, 7, 19, 39]. While Lai et al. [19] and Eilertsen et al. [7] must balance the data fidelity and temporal consistency in the process of training on a large dataset, it is much easier for us to achieve this goal since our method is trained on a single video. Fig. 9 shows that data fidelity and temporal inconsistency are both increasing in the process of training.

In principle, the training epochs should be different for videos with different duration. For example, a video with 200 frames requires fewer epochs than a video with 50 frames. For two videos with the same length, the video with smaller motion requires fewer epochs. In our experiment, we observe that temporal consistency is stable in many epochs. For example, $E_{warp}$ is only increased by around 0.002 from 25-th to 80-th epoch in Fig. 9. Therefore, we simply select the same epoch (25 or 50

epochs) for all the videos with different length (30 to 200 frames) in a task based on a small validation set up to 5 videos. We do not need to carefully select the epoch because reconstructing the flickering artifacts takes much more time compared with common video contents.

**Computational cost.** For an $800 \times 480$ video frame, our approach costs 80 ms for each iteration during training on Nvidia RTX 2080 Ti. For a video with 50 frames, 25 epochs (1,250 iterations) cost about 100 seconds for training and inference. In this case, the average time cost for each frame is 2 seconds. Note that U-Net is replaceable in our framework, and a lightweight CNN may speed up the whole process.

## 5 Discussion

We have presented a simple and general approach to improving temporal consistency for videos processed by image operators. Utilizing the deep video prior that the outputs of CNN for corresponding patches in video frames should be consistent, we achieve temporal consistency by training a CNN from scratch on a single video. Our approach is considerably simpler than previous work and produces satisfying results with better temporal consistency. Our iteratively reweighted training strategy also solves the challenging multimodal inconsistency well. We believe that the simplicity and effectiveness of the presented approach can transfer image processing algorithms to its video processing counterpart. Consequently, we can benefit from the latest image processing algorithms by applying them to videos directly.

One of the limitations of our method is the relatively long processing time at the test time. Although we do not need to train on a large dataset, we need to train an individual model for each video, which costs more time than direct inference compared with Lai et al. [19]. Nevertheless, unlike previous approaches that explicitly adopt optical flow to enforce temporal consistency, we demonstrate this video prior (i.e., temporal consistency) can be achieved implicitly through the neural network training.

In the future, we will focus on improving the efficiency to shorten the processing time of application in practice. Also, we believe the idea of DVP can be further expanded to other types of data, such as 3D data and multi-view images. DVP does not rely on the ordering of video frames and should be naturally applicable to maintaining multi-view consistency among multiple images. For 3D volumetric data, 3D CNN may also exhibit a similar property of DVP.

## Broader Impact

With our proposed framework, users can apply numerous existing image processing and enhancement algorithms to videos with strong temporal consistency. As a consequence, instead of paying extra efforts to solve the temporal inconsistency problem, researchers can spend more time utilizing video information for improving the performance. Moreover, the simplicity of our approach promotes the potential of wide deployment. Our proposed *Deep Video Prior* is an implicit characteristic of a CNN training on a video and can be used to replace a complicated handcrafted regularization term. We believe this observation can inspire researchers to utilize CNNs for video processing tasks better.

## Acknowledgments and Disclosure of Funding

We would like to thank the anonymous reviewers for their constructive comments. This work was completed in part with resources supported by the Hong Kong University of Science and Technology.

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
