[Reviews · NeurIPS 2020]

Review 1

Summary and Contributions: In this paper, the authors borrow the idea from deep image prior and extend it to blind temporal video consistency. Rather than training on large scale video datasets and relying on explicit temporal regularization, this paper proposes to train one network for few epochs only on the target video. To demonstrate its effectiveness, seven different tasks are tried and show superior performance.

Strengths: 1. Though not that novel, extending the deep image prior to deep video prior sounds interesting. Empirically, it makes senses since the network learning processing is often coarse to fine. 2. This method does not require massive video data for training and only need to train the network on the target video pair. 3. Seven different computer vision tasks are tried and demonstrate the generalization ability to some extent.

Weaknesses: 1. According to the underlying hypothesis of this method, I guess the proposed method will not work well and produce the averaged effect if the processed video {P_t } has very bad temporal consistency (similar pixels have very different values along the temporal dimension). The original neural style transfer with random initialization for each frame should be a typical case. As shown in the supplementary video, the cyclegan result at 00:48 is more blurry than the original processed one, and all the red color in the colorization result at 00:54 disappears. In other words, this method can only handle slightly temporal inconsistent video. 2. Since a new network is used to approximate the behavior of the original image f, it may significantly degrade the original performance of f. For example, for dehazing, the temporally smoothed results may have much worse performance than original dehazing results. However, the authors have not provided such experiments. Considering many of these tasks have their own quantitative evaluation metrics, I highly recommend the authors to evaluate the processed videos with the original task-specific metrics. 3. The overall idea is very similar to deep image prior and just a extension, so it is not that new to me. 4. The proposed method need to train one specific network for one target video, even though some videos share the same f. 5. As shown in the analysis in Fig 8, the temporal inconsistency and the perceptual similarity both increase, which means that we need to tune the training epochs for each video or task very carefully. In this sense, I do not think it is easier than [6,17]. In other words, to achieve satisfactory temporal consistency, the original performance of f will degrade significantly.

Correctness: Please refer to the weakness part.

Clarity: The writing is overall good.

Relation to Prior Work: Overall satisfactory, but it misses many highly related works in video style transfer. In fact, the evaluation formulation in (6) is originally proposed from such methods. Typical examples include: Artistic style transfer for videos Coherent Online Video Style Transfer Real-time neural style transfer for videos

Reproducibility: Yes

Additional Feedback: **************** after rebuttal *************** According to the response, I have some feeling that the authors are not confident to give the performance, and I guess this method will make the results of the original task much worse. Another concern is about handling the large temporal inconsistency, I cannot buy the argument the authors made in the response. I double checked Table 2, Table 3, Fig. 5, and the supplement, but do not think these are the cases I mentioned. So I will stick on my first judgement that this method cannot handle the large temporal inconsistency. Considering all other reviewers are pretty positive, I am okay if this paper is finally accepted, but the authors must provide the performance influence analysis in the final version and improved the texts based on the above suggestions.


Review 2

Summary and Contributions: This paper proposes a new method for blind temporal consistency, namely Deep Video Prior. The main idea is to train a CNN (on the testing video at test time) to learn the video prior (prior of this testing video, aka exemplar-video-prior) while enforce the consistency between processed frames and original frames. IRT is also proposed to handle multimodal inconsistency. Experiments are solid compared with previous methods [3, 17] on quantitative metrics, qualitative results, as well as on a user study. Written presentation is clear and easy to understand.

Strengths: - The proposed deep video prior (DVP) is novel and interesting. - Strong experimental results outperforming previous work [3,17] on various tasks. - Good ablation results, the experiment to motivate the proposed IRT makes sense. - Finally, I like the the fact that the paper does not stop at some normal metrics, but go further with user study. This makes the evaluation more complete and convincing.

Weaknesses: - DVP needs to perform training at test time (25-50 epochs) per testing sequence. - The reviewer understands that the Figure 8 provides some insights "when to stop", however, it is unclear how it will change or is it sensitive to the length of videos (longer videos). - It is interesting to see how DVP perform on video with different length?

Correctness: I think most of the claims made in this paper are correct.

Clarity: The paper is well written and organized easy to understand.

Relation to Prior Work: The paper covers enough previous work in my opinion.

Reproducibility: Yes

Additional Feedback: - It may be interesting to test this approach on a 3D CNN, the temporal consistency may be stronger. One example architecture can be V2V [a] or similar U-Net style 3D CNN. V2V is a little bit outdated e.g. no residual connections. - The idea in this paper is also related to distillation, here DVP distill from image model to video data at testing. Of course training is lightweight since it trains only on 1 testing video. It may be worth to briefly mention in related work. [a] Tran et al. Deep End2End Voxel2Voxel Prediction. CVPR'16 Deep Learning workshop. === after rebuttal === I adjust my rating to 6 due to the following reasons: 1) I still think the submission has enough novelty and technical contribution (compared with DIP) and worth publication. That's why I am still voting for accepting this paper. 2) However, I do agree with R1 that the paper need to report the performance on original metrics and compared them with processed frames and other baselines. This is normally doable during the rebuttal period since it requires no additional training/fine-tuning. However, the authors did not provide these comparison on their rebuttal. If the paper is accepted, I recommend the authors should include those comparison to the camera-ready even if it does not outperform processed frames and/or other baselines. It needs to give the readers the full/complete view of the proposed method. The authors may need to give explanation/insights why the proposed method degrades/trades the original metric for the perceptional coherence in video.


Review 3

Summary and Contributions: This paper proposes an algorithm to improve the video temporal consistency based on a deep video prior. The proposed method learns to reconstruct a test video and adopts an iteratively re-weighting strategy to solve the multi-modal inconsistency problem effectively. The visual results in the supp video show better short-term and long-term temporal consistency than existing approaches.

Strengths: - The idea to learn a deep video prior for blind temporal consistency is interesting and novel. - The paper is well written and easy to understand. - The quantitative and qualitative results are clearly state-of-the-art.

Weaknesses: - Some technical details are not clear enough and require more elaboration. For example, in the iteratively reweighted training (sec 3.2), it's not clear how to generate two different frames O_t^main and O_t^minor from the same input I_t at the same time. From the evaluated tasks used in Section 4, all these models are deterministic after training, which means that the model will generate the same result from the same input frame. It's possible that a model can generate very different results from the same video (as shown in the second example in Figure 5). But for the same input frame, the output should be the same unless the authors have trained two different models. Could the authors explain a bit more on this? - From line 244, the number of training epochs is determined by the authors for "each task". Could the authors provide some guideline about how to choose this number? - The perceptual metric (8) computes the PSNR between P_t and O_t. However, it's already known that PSNR does not correlate to human visual perception well. Instead, the VGG loss or LPIPS score is commonly used for evaluating the perceptual quality.

Correctness: Yes.

Clarity: Yes.

Relation to Prior Work: Yes.

Reproducibility: Yes

Additional Feedback: Please see the weaknesses.


Review 4

Summary and Contributions: This paper proposes to apply “Deep Video Prior” for blind temporal video consistency (i.e., removing temporal flickering from a video processed by an unknown image-based algorithm). Unlike existing methods that explicitly use optical flow as part of the training loss [Lai et al. 2017] or objective function [Bonneel et al. 2015], the proposed method trains a randomly initialized CNN so that its outputs match the processed video for each frame. The core idea is that it’s easier for a CNN to fit the contents of the processed video than fitting the temporal flicker. As a result, one can recover the temporally consistent video by training a particular number of epoch (before the model overfits to the temporal flicker). The paper also introduces a method for handling the multimodal inconsistency problem. The core contribution of this paper lies in a new application of deep image prior to the time dimension.

Strengths: **Exposition** - The paper is generally well-written. I appreciate the authors’ toy example to help the reader understand the concept of uni-/multimodal results. **Novelty** - This paper extends the idea of Deep Image Prior (DIP) to videos. It applies DIP to an interesting problem context (blind video temporal consistency). I feel that it’s not entirely precise video prior as the CNN is still trained to fit one image at a time. However, I like the insight that fitting the temporal flicker of a processed video takes a longer time than fitting the contents. - The paper also addresses a problem of multimodal results using the proposed iterative reweighted training strategy. **Method** - The method is simple and does not require external training data. While it runs slower than other learning-based methods, it’s still interesting. I can imagine that this paper would inspire future research on exploiting deep video prior to other problem. **Evaluation** - The evaluation is fairly comprehensive (comparing with two baselines on 7 computer vision tasks). User study suggests that the results are more preferable by users. - The code will be made available.

Weaknesses: **Exposition** - There are several places where the exposition is not clear (see the “clarity section” below). **Novelty** - One could argue that this paper is an application of DIP to video. Yet, I think the insight is interesting and differs from recovering clean images in the original DIP paper. **Method** - When I see that the title of deep video prior, I thought there will be some specific design exploiting the temporal dimension of video. Instead, the method applies the CNN training for each frame independently. I am wondering whether alternative designs could be applicable for the problem of blind video consistency as well. For example, there could two other simple methods to consider. - 1) One can have a straightforward extension of DIP for video by using some kind of 3D (spatio-temporal) CNN. The same argument that fitting the temporal flicker takes longer than fitting the contents of the processed video still applies. - 2) Explicitly using flow as in the “An internal learning approach to video inpainting, ICCV 2019” paper. - The proposed method requires 2 seconds per frame for frames with 800 x 480 resolution on a NVIDIA RTX 2080 Ti. I think it would be better to acknowledge slow runtime as a limitation by comparing the runtime speed with other methods. I don’t know how other methods’ runtime performances are in this specific hardware setup. However, from [Lai et al. 2018], “…. the execution speed achieves 418 FPS on (Titan X) GPU for videos with a resolution of 1280 × 720”. This suggests that the proposed method may be around 1,000 times slower than that of [Lai et al. 2018]. **Evaluation** - As discussed in the paper, the perceptual similarity and warping error are competing metrics. Showing one data point cannot reveal the whole story. I would suggest showing a *curve* in the perceptual similarity and warping error space. For example, one can plot the two errors along the training iterations. - Why using PSNR as the perceptual similarity? It does not reflect the perceptual similarity between the predicted and processed frames. I think the metric used in [Lai et al. 2018] (LPIPS) makes more sense here. - I have concerns about “L244: In this paper, we fix the number of training epochs for each task empirically.” How did you determine the number of training epochs? As there is no mention of a validation set in the paper, it sounds like this is tuned on the *testing set*. What are these numbers for different tasks? Are they very different (and therefore sensitive) to different tasks. This is important because the goal for this problem is to enforce temporal consistency on videos that are processed by *unseen* application. That is, the method should work without the knowledge of accessing the information about the task. If one needs to tune the hyperparameters for each task, then it is no longer a “blind” video temporal consistency method. - There were failure cases shown in the paper. It would be good to have some to highlight the limitations of the method.

Correctness: Yes, I believe that the claims and the methodology are correct.

Clarity: The paper is well written. The figures are informative. The method exposition is sufficiently clear. There are a couple of places that could be further improved. **Figure 2** - The left figure shows that at test time, the method in [Lai et al. 2018] uses the test video {I_t}_{t=1}^T as input. This is not accurate. Instead, it takes the processed video {P_t} and the input video {I_t} as input and produces the output video {O_t}. Also, the figure suggests that the method produces two frame outputs (with two yellow borders). This is also not accurate as it produces one output frame at a time in an online fashion (as opposed to the offline approach in the paper). **The paragraph “Correspondence-based regularization” (starting from L72)** - “avoid complicated optical flow estimation”. As there are many off-the-shelf and accurate flow estimation algorithms available, I think characterizing optical flow estimation as “complicated” may be a bit misleading for the readers. - The method in [Lai et al. 2018] applies optical flow only in the training stage, it does not need to estimate flow at test time. **Section 3.2 Iterative reweighted training** - The exposition of this section should be improved. - 1) The paper states that the ideas are from IRLS and K-means. However, no connections to these approaches given. For example, it would be nice to connect the proposed method to the “cluster assignment step” and the “cluster center update step” in the K-means algorithm. - 2) Notation should be clear. Examples: o Specify the dimensionality of C_t. o Use a notation for “elementwise multiplication” between “C_t P_t” and “C_t g(I_t, \theta^i )”. o Shouldn’t the confidence map C_t also depend on the number of iteration i? o In Eqn 4, clarify that C_t is C_t^{main} or C_t^{minor}. - 3) “if a pixel x is closer to O_t^{main}” is not precise. In what space do you measure the distance? From the sentence, it seems to refer to “spatial distance” (which is not correct). - 4) Eqn 5 is confusing. What does this equation do? Please provide the intuition of the design of this. - 5) Where does the O_t^{minor} come from? Do we train the same CNN to generate these two frames at the same time? Or do we train two separate CNN? How do we train that model to produce O_t^{minor}?

Relation to Prior Work: Yes, the authors did a good job providing an overview of the line of works on video blind temporal consistency (L24-L35). I think the paper should include more discussions with [A]. The method in [A] also uses “deep video prior” for the task for video completion. It would be good to discuss the similarity/difference with this paper in the introduction section (e.g., on L48-L51). For example, the method in [A] uses optical flow explicitly while the proposed method does not. What are the advantages/disadvantages of using explicit flow information? [A] Zhang, Haotian, Long Mai, Ning Xu, Zhaowen Wang, John Collomosse, and Hailin Jin. "An internal learning approach to video inpainting." In Proceedings of the IEEE International Conference on Computer Vision

Reproducibility: Yes

Additional Feedback: Reference errors: [25] The paper has an only single author. Please remove the “et al.” [27] The rest of the references use full names for the authors. Please include full author names for consistency. == Post-rebuttal comments == I read the authors' rebuttal and comments from other reviewers. I am still leaning to accept. Here are some of my thoughts: 1) Novelty: - I like the idea of applying DIP to the temporal consistency problem. I feel that the idea is fairly interesting and it would not be fair to describe this idea as a simple extension. The iteratively reweighted training is also interesting and has not been explicitly addressed in prior work (although the exposition can be further improved). 2) Experimental results: *Handling large temporal inconsistency*: - I guess the authors' response is trying to say that their experiment (Table 2, Table 3, and Figure 5) did show their method outperforms (or at least competitive) with respect to the two baseline methods. I completely agree with that Dongdong that the method may not be able to handle videos with large temporal inconsistency (e.g., video completion problems). However, as far as I know, existing blind temporal consistency methods also suffer from such limitations. * Performance degradation on the original task*: - I think this is a great suggestion and indeed the authors did not provide such results in the paper and in the rebuttal. A "proxy" of this would be S_perceptual in Table 2 (last column). As the task is blind to the image-based method, we usually do not expect to outperform the original algorithm so the goal here would be to maintain the original content as much as possible. I think it would be nice to include these results in the supplementary material, but I don't think the outcome of the result would change my assessment of the paper. That being said, I do have multiple complaints about the paper, including the unclear exposition, slow speed, missing baseline 3D CNN based methods, missing perceptual error (e.g., LPIPS) as metrics, and selection of training epochs. So, in summary, - At some fundamental level, I believe the paper is good. It takes a different route to the video temporal consistency problem and show its promise. I think this will inspire more future work. - There are quite a few major changes that I would like to see in order to be happy with the paper (most of the suggestions in my and others' reviews)

[Author Response · NeurIPS 2020]

We thank the reviewers for their constructive comments on our paper. We address the major questions in the following.

**R1: The ability to handle large temporal inconsistency.** Our approach can handle large temporal inconsistency such
as multimodal inconsistency with the IRT strategy and outperform baselines, as shown in Table 2, Table 3, Fig. 5, and
the supplement. In the experiments for most evaluated tasks, we do not observe the extreme cases mentioned by R1. As
for the extreme case, we believe no existing blind temporal consistency approach can perform well. In the colorization
example, note that red color is an inconsistent artifact and is thus learned as a minor case (see Fig. 6). We do not observe
obvious blurry artifact in the mentioned CycleGAN case and users prefer our results compared with the original video.

**R1: The performance degradation problem.** Avoiding performance degradation is critical to blind temporal consis-
tency, and perceptual similarity is used to analyze performance degradation. Table 2, Table 3, Fig. 5, and the supplement
show quantitative and qualitative comparisons. For dehazing, the comparison with original videos is provided at $0:13$
in the supplement and our performance is not degraded perceptually. The user study takes both perceptual preservation
and temporal consistency into account, and the preference rate of our method is much higher than that of baselines.

**R1: Just an extension of DIP?** Our DVP is not a direct extension of DIP, and there are substantiate differences in
several aspects (1) Implementation. DIP reconstructs the image from noise while DVP tries to learn the mapping
from input frames to processed frames. A simple extension of DIP on video should be: reconstructing the video from
noise by a 3D-CNN. (2) Assumption. DIP assumes image prior exists in CNN architectures; DVP assumes video
consistency enforced by correspondences can be learned from the internal similarity of frames. (3) Application. DVP
can enable numerous image processing methods applicable to videos while maintaining temporal consistency. (4)
IRT. The proposed IRT solves the multimodal inconsistency problem well, which is ignored by prior work. We treat
flickering artifacts of unimodal inconsistency as noises in the temporal domain, which is an important observation.

**R1, R2, R4: Train a specific network for each video and running time.** Training on a test video takes about 2
seconds per frame, which is not real-time. Moreover, the comparison of 1000 times is not reasonable since test
environment is different. Compared with direct inference, extra 24 / 49 epochs are required for training. Besides, we
can try to speed up the model by using a lighter model. Also, our approach has advantages: no need for training on a
large dataset, which may take hours or even days; domain gap between a training set and a test set does not exist.

**R1, R2, R3, R4: Carefully tuning? Relationship with the length of videos. How to select the epochs? Is it still
"Blind"?** (1) One basic observation is that reconstructing the flickering artifacts takes **much more** time compared
with common video contents. For example, in Fig. 9, $E_{warp}$ is only increased by 0.002 from 25-th to 80-th epoch.
Hence, we do not need to tune the epochs carefully. (2) Since the network learns a temporal consistent image mapping,
the training time is decided by the iterations. For the same kind of video, a video with 200 frames requires fewer epochs
than a video with 50 frames. It is an interesting idea to further study the relationship between video length and the
number of epochs needed. (3) In the experiments, since the temporal consistency is great in many epochs, we simply
select the same epoch (25 or 50 epochs) for all the videos (30 to 200 frames) in a task based on a small validation set up
to 5 videos. (4) Our approach is blind because we always treat every image operator as a black box.

**R1, R3, R4: Other metrics, e.g., LPIPS, VGG loss or original metrics.** We use PSNR because PSNR is a common
metric for data fidelity in many tasks we evaluate. We also have a user study to evaluate human perceptual preference.
We agree that LPIPS and VGG loss are good metrics for perceptual similarity, and we will add VGG loss and LPIPS in
the final version. Fig. 9 should be the curve of perceptual similarity and temporal inconsistency mentioned by R4.

**R2, R4: 3D CNN, distillation and optical flow.** Great advice on more comparisons. (1) Reconstructing a video from
noises by a 3D CNN can be a simple extension of DIP. Hence, we believe such a 3D CNN model is memory hungry, and
we will analyze it. Also, the multimodal problem can be a challenge in this simple method. (2) Our method works for
both learning and non-learning based operators. If the image operator is a CNN, it is similar to train a student network
from the teacher network on a single video. We believe we can adopt a lighter CNN architecture to speed up. (3) Using
optical flow is often useful for short-term temporal consistency. However, optical flow is usually not accurate enough
for long-term consistency (Line 34-35, caption in Fig. 5), and more comparison is described at Line 72-87.

**R3, R4: Clarification for IRT in Sec. 3.2.** With IRT, we increase the number of channels in the network output (e.g.,
six channels for two RGB images). Then, in each iteration, Eq. 5 is used to divide pixels into two clusters (main and
minor modes). This is similar to cluster assignment in K-Means when K=2 and the pixels in minor mode are similar
to the outliers in IRLS. We use $L1$ distance as spatial distance $d$, as mentioned in Line 146-147. Then pixels in two
clusters are used for updating two modes in each iteration. The two modes are generally different in different iterations
since they are obtained from different pixels. At last, notations will be revised, thank you.

**R4: Discussion and exposition.** We will add discussion and revise Fig. 2, the paragraph from Line 272, and Sec. 3.2.

**R1, R2, R4: Additional related work.** R1: Yes, these methods use the metric and we will include them. R2: We will
analyze these works and add them. R4: Yes, this work also uses some type of video prior and we will discuss it.

[Meta-Review · NeurIPS 2020]

Four experts reviewed the paper. Three of them placed the paper above the acceptance threshold, and one put it marginally below. The reviewers were happy with the rebuttal in general, but they all wondered about the original task's results. Based on the reviewers' feedback, the decision is to recommend the paper for acceptance. The authors are encouraged to address the reviewers' questions, especially about the original task's performance, to the best of their ability when preparing the camera-ready. We congratulate the authors on the acceptance of their paper!